# Availability Assessment of IMA System Based on Model-Based Safety Analysis Using AltaRica 3.0

**Haiyong Dong [1,*,†]**, **Qingfan Gu [2,†]**, **Guoqing Wang [1,2,3,†]**, **Zhengjun Zhai [1,†]**, **Yanhong Lu [1,†]** and **Miao Wang [3,†]**

1  School of Computer Science, Northwestern Polytechnical University, Xi'an 710072, China; wang_guoqing@careri.com (G.W.); zhaizjun@nwpu.edu.cn (Z.Z.); yanhonglu@nwpu.edu.cn (Y.L.)
2  China National Aeronautical Radio Electronics Research Institute, Shanghai 200233, China; gu_qingfan@careri.com
3  School of Aeronautics and Astronautics, Shanghai Jiao Tong University, Shanghai 200240, China; wang_miao@careri.com
*  Correspondence: donghaiyong@mail.nwpu.edu.cn; Tel.: +86-182-2050-7569
†  These authors contributed equally to this work.

**Abstract:** The integrated modular avionics (IMA) system is widely used in all classes of aircraft as a result of its high functional integration and resource utilization in developing advanced avionics systems. However, a series of challenges related to safety assessment exist in the background of the logical architecture for multi-message interactions of the IMA system. Traditional safety assessment methods are mainly based on engineering experience, and are difficult to reuse, incomplete, and even error-prone. Here we propose a method to assess the availability of the IMA system based on the thinking of model-based safety analysis. To aid the proposed method, we implement a tool to generate a AltaRica 3.0 file used to assess the IMA system model. The simulation results show that the proposed method makes the availability assessment fast, efficient, and effective. Moreover, we apply this method to the modification analysis of the IMA system under the condition of satisfying the safety requirement. Our study can enhance the safety assessment of safety-critical systems effectively, assist the design of IMA systems, and reduce the amount of errors during the programming process of the safety model.

**Keywords:** availability assessment; integrated modular avionics; model-based safety analysis; AltaRica 3.0

## 1. Introduction

Integrated modular avionics (IMA) is the state-of-the-art methodology in the real-time computer network airborne system domain, which consists of a number of computing modules capable of supporting numerous hosted applications with different criticality levels [1,2]. Up to now, IMA has been widely used in large, civil aircraft, such as the Airbus A380 and Boeing B787, due to the remarkable improvement in system efficiency, with weight and power consumption reductions by means of comprehensive resources integration or high resources sharing [3]. Different from the federated digital architecture, the IMA system can be divided into three levels: the functional layer, logical layer, and physical layer. The visual objects in the logical layer work together to provide services for hosted applications in the functional layer by utilizing the resources in the physical layer. In addition, some IMA systems, like that in the A380, use two redundant avionics full duplex switched ethernet (AFDX) networks to guarantee the required availability [4]. However, at the same time, the reuse of the traditional safety assessment will become more difficult. Virtual link (VL), the central

feature of an AFDX network, is a unidirectional logic path from the source end-system to all the destination end-systems [5]. In this way, VLs are mapped onto visual objects in the logical layer, AFDX switches and end-systems are mapped onto the resource in the physical layer, and functions are mapped onto the application in the functional layer. In practice, the system engineer utilizes the IMA configure tool to obtain a specific VL configuration, whose network performance meets the needs of hosted applications. However, the above VL configuration needs to be further analyzed to verify that the availability of specific applications appropriate to a required criticality level is satisfied. Availability is the qualitative or quantitative attribute that signals that a system is in a functioning state at a given point in time, and it is sometimes expressed in terms of the probability that a system does not provide its output(s) (i.e., unavailability) [6]. It is an important factor in the area of reliability and safety, especially for the safety-critical system. Traditionally, the safety assessment and hazard analysis are modeled on fault tree analysis (FTA) by analysts based on engineering experience, which is easy to understand, but hard to reflect in real designs. Even more important for complex avionics systems, the FTA model is too huge to modify with any minor change by manual operation [7]. In addition, traditional safety analyses (FTA, etc.) are usually based on informal system models, which are always regarded as incomplete, inconsistent, and error-prone [8]. Moreover, a consistent formal model is needed in both system design procedure and safety analysis procedure. To solve these problems, model-based safety analysis (MBSA) is proposed.

　　Up to now, MBSA has been widely used in the fields of aviation [9], railways [10], automotives [11], and other safety-critical systems [12]. During the process of MBSA, system engineers and safety analysts share a common system model. It extends the system model with a fault model as well as relevant portions of the physical system, and is recommended to model complex systems in ARP 4761A draft [13]. In addition, Laboratoire Bordelais de Recherche en Informatique (LaBRI) developed a free formal language, AltaRica, to model both functional and dysfunctional behaviors of systems. Models in AltaRica 3.0 are described by guarded transition systems (GTS), which consists of state variables, flow variables, events, transitions, and assertions [14]. AltaRica 3.0 can support the modeling of event driven systems based on MBSA, and the model described can be hierarchical and compositional [15]. Thus, AltaRica 3.0 has been widely used to model these safety-critical systems [16,17].

　　Some researchers have investigated the safety assessment of avionics systems based on MBSA. Morel used MBSA to validate several IMA architectures with three levels, and suggested that MBSA is a good method for safety assessment in early validation to support flexible and rapid prototyping of integrated systems, and expressed that his study needed to do some quantity analysis to verify whether the availability further met the requirements [9]. Li used MBSA to study the safety assessment of complex aircraft products, proposed a safety modeling approach based on AltaRica, and proved its validity through simple hydraulic system verification [18]. The safety analysis of IMA based on MBSA have also been studied, while the model described by AltaRica was totally coded by hand, this makes it difficult to reuse and easy to make mistakes with [19].

　　In this paper, to study the impact of using the effective procedure and tool to analyze the safety of IMA systems, a method based on MBSA using AltaRica 3.0 to assess the availability of the IMA system is proposed and a tool to aid the assessment method is implemented. An IMA system case is modeled to verify the validity of the proposed method. In addition, we do some research on design optimization of the IMA system. Finally, the advantages and disadvantages of the different assessment methods are analyzed. This provides new insights into the safety assessment and hazard analysis in an IMA system operating within an acceptable safety level.

## 2. Assessment Method

　　Model-based safety analysis (MBSA) is able to build a complete, accurate, and consistent safety model for complex, safety-critical systems [20]. Generally, there are seven steps in a MBSA process: "Gather the most complete system data available at the time", "Define the goal and the granularity of the analysis", "Define the failure conditions to be studied", "Build the failure propagation model

(FPM) according to the collected data", "Build the failure condition logic", "Verification of the FPM and failure condition logic", and "Failure condition evaluation & analysis". However, not every MBSA process suits the above steps. In addition, there is no limit to the model languages in a MBSA process. Considering this, AltaRica 3.0, an available, high-level language for event driven modeling of complex systems, is especially well suited for safety analyses and performance analyses. AltaRica 3.0 defines the block by representing the component with failure mode, which is composed of the declaration of variables and events, and the definition of transitions and assertions [18].

As shown in Figure 1, the system designer is responsible for the system model while the safety engineer is responsible for the safety model. An IMA system model consists of three layers: the physical layer, logical layer, and functional layer. These three layers have one-to-one correspondence with the failure modes, failure propagation models, and failure conditions, which constitute an IMA safety model. According to the requirement, the system designer utilizes the IMA configure tool to generate the xml file for data exchange. The file in xml format contains the failure rate and configured VLs of every component to describe the logical relationship between the components in the IMA system. In this paper, the data about safety of failure modes and failure propagation models can be collected from the system model and translated into the description of the alt file, which is used to describe the safety model. The safety engineer obtains the information about the function from the system designer and defines the invalid function as the studied failure condition. The configuration xml file is used as interface control document (ICD) for components in IMA system. Note that the ICD file is built by the system designer, and it is always regarded as the input for the safety engineer.

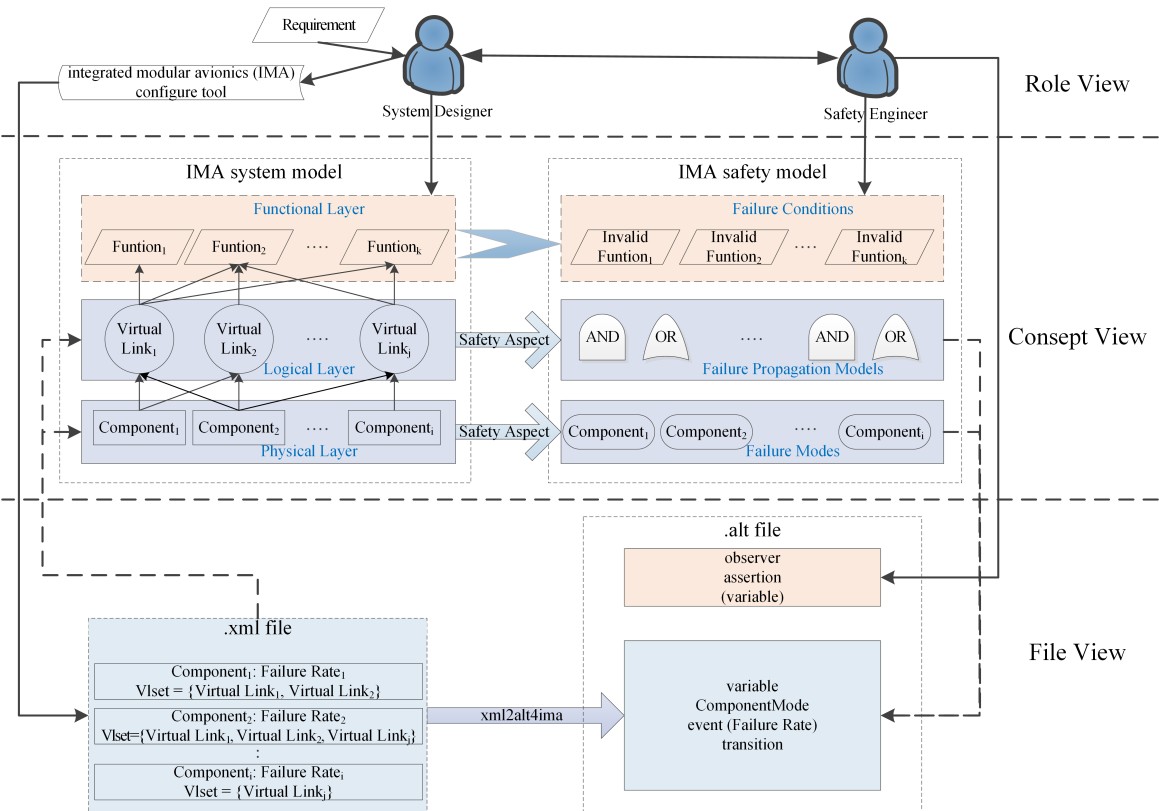

**Figure 1.** Generated mapping from system model to safety model for a integrated modular avionics (IMA) system

In this paper, a method is proposed to assess the availability of the typical IMA system based on MBSA using AltaRica 3.0. On the basis of the system model described in the xml file, we build the safety model in the alt file (AltaRica 3.0 format) by utilizing a tool named xml2alt4ima which was developed by our team. In the xml file, there is one root element named "VirtualLinks" after XML

declaration. Each component is an element of the VirtualLinks. Every component has three attributes to represent "Name", "GuID", and failure rate of "Loss". Every component also contains several elements to represent the configured VLs. Every VL consists of 10 attributes: "Name", "GuID", "BAG", "MaxFrameSize", "MinFrameSize", "Priority", "Captain", "ActualPath", "Source", and "Destination". Note that not every attribute is utilized in our method, for example, "BAG" is used to set the time gap between two packages, and it's used to assess the performance in real-time. In our method, "Name" and failure rate of "Loss" of each component, and "Name" and "ActualPath" of each VL are the safety properties and they are utilized in describing the safety model.

The xml file contains all components with thier configurations, and these configurations can be mapped onto VLs in the physical layer and components in the logical layer. In addition, since the fault may occur in every component in the physical layer, each component has a responding failure mode in safety model. Each VL in the logical layer has its own working status, which can be changed when fault occurs in the related components. The failure propagation models and failure modes are described by "variable", "ComponentMode", "event", and "transition" in the alt file. The "variable" is used to describe the state of the system or subsystem, the "ComponentMode" is used to describe the status of component, the "event" is used to describe the event that may occur in the system, and the "transition" describes how the system evolves. The variable with brackets in the alt file generated by the functions means it is not required, but it can assist in the process of calculating the observer, especially when the observer is a complex function or failure condition. Moreover, the functions in the functional layer are defined by the system designer, and it is the basis of the failure condition in the safety model during the process of MBSA. The failure condition is a condition having an effect on the aircraft, which is usually caused by one or more failures or errors associated with the flight phase, relevant adverse operational or environmental conditions, or external events [21]. To sum up, the system model described in the xml file can be mapped onto the physical layer and logical layer, and it contains the basis data of the safety model which is described in the alt file. The safety engineer needs to understand the system model, extract valid information of the safety assessment, build the failure condition with the help of the system designer, and thus, realize the safety model.

Therefore, the process to assess the availability of the IMA system can be concluded in a method as follows:

- **Step 1. Define the failure condition of the IMA system and their safety requirements.**
  The failure condition means an unexpected state. It is always a logical combination of some unexpected states. For the IMA system, it means an invalid function.
- **Step 2. Utilize the special generation tool (xml2alt4ima) to generate an alt file based on the configuration xml file.**
  The xml2alt4ima tool is designed to aid the construction of the alt file according to the xml configure file.
- **Step 3. Manually add the observer, assertion, and variables if needed.**
  Complete the alt file manually. The observer is used to represent the failure condition and complex function. The assertion contains some sentences to represent the logical relationship. Variables provide assistance in understanding the logical relationship between the failure condition and failure mode. In addition, we need to add a variable named "failed", which is used to represent the top event of the fault tree.
- **Step 4. Utilize the AltaRica 3.0 assessment tool to compile the alt file, and obtain the cut set, probability, contribution, and so on.**
  The AltaRica 3.0 compiler can explain the meaning of the alt file. We recommend the free OpenAltaRica tool [22], which integrates many analysis functions.

The xml2alt4ima tool is developed in Matlab 2016a, and the core algorithm is illustrated in Algorithm 1.

---

**Algorithm 1** The algorithm to generate the alt file from the configuration xml file of the IMA system.

---

**Input:** xml file, including $m$ ($m \geq 1$) components with failure rate and configured VLs;
**Output:** alt file, including file structure, event, transition and required variable;
1: Begin initialization
2:     Define the domain of ComponentMode for all components
3: End initialization
4: $m \Leftarrow$ the quantity of the components in the xml file
5: For component $i$ ($1 \leq i \leq m$)
6:     $Rate_i \Leftarrow$ the failure rate of component $i$
7:     Define the state of component $i$ based on ComponentMode
8:     Define the event for component $i$ with $Rate_i$
9:     Define the transition of component $i$ based on event
10:     n $\Leftarrow$ the quantity of the VLs configured in component $i$
11:     For VL $j$ ($1 \leq j \leq n$)
12:         Define variables for VL $j$ configured in component $i$
13:         $p \Leftarrow$ he quantity of components in the actual path through VL $j$
14:         For component $k$ ($1 \leq k \leq p$)
15:             Add action for VL $j$ in the transition configured in component $i$
16:         End component $k$
17:     End VL $j$
18: End component $i$
19: Delete redundant variables for VL
20: Begin modification
21:     Add assertion for the failure condition
22:     Add block for the whole model
23: End modification

---

## 3. Case Study

In this section, a typical example of the IMA system model is introduced in Section 3.1, some general assumptions and failure condition are presented in Section 3.2, the results based on our proposed method is calculated, and it is also verified by other methods in Section 3.3. On the basis of the results, we try to optimize the system model and propose advice for the system designer in Section 3.4. We also try to explore the efficiency of different safety assessment methods in Section 3.5.

### 3.1. IMA System Model

As a result of the high requirement of performance and availability, the utilization of the existing resource becomes the most difficult point in the structural design of IMA and the core architecture of civil avionics systems [23,24]. For example, to avoid a single-point failure, all AFDX networks and end-systems are designed to be double or triple module redundant. Figure 2 shows a typical IMA system model with two redundant AFDX networks, three general processing modules (GPM), three remote data concentrators (RDC), and two hosted functions (HF). The RDC is designed for data acquisition from the sensor (SEN) and other signal sources, the GPM is designed for data calculation and procession, the HF is designed for data display and upper application, and the switch (SW) is designed for transferring data through the IMA system. It is assumed that HF1 is used by the captain and HF2 is used by the copilot. Every HF needs data processed by the GPM from both SEN1 and SEN2.

SEN1 BU denotes the backup of SEN1, and SEN2 BU denotes the backup of SEN2. Every RDC obtains the data from the connected sensors through the ARINC 429 bus, and transfers these data to three GPMs through the redundant network. After processing these data, every GPM transfers the processed data to two HFs through the redundant network. The GPM and HF are able to utilize the

effective data and drop the redundant data. In addition, since the sensors are connected with the RDC through the ARINC 429 bus, instead of the AFDX switch, there is no VL configuration in the sensors.

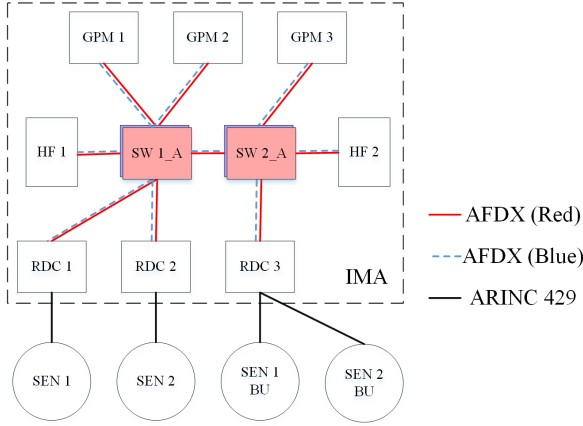

**Figure 2.** The model of a integrated modular avionics (IMA) system

### 3.2. Assumption and Failure Definition

There are 12 components in the IMA system model, as shown in Figure 2. To simplify the model, there are six general assumptions [24–26].

**Assumptions**

- Faults are modeled as statistically independent distributed events;
- The failure rate of each component is a constant;
- A fault occurs instantaneously and at most one fault event in a minimum time slice;
- The system and its components have two states: normal and failure;
- The system and its components are unrepairable while in use;
- The cable between two components keeps working.

Note that the failure distribution of the components is assumed to be a $\lambda$-exponential distribution where $\lambda$ is equal to the failure rate per flight hour. The mean time between failure (MTBF) and failure rate per flight hour of these components are shown in Table 1. The failure rate of the sensor comes from the book written by Jukes [27], the MTBF of the switch and GPM come from Reference [4], and the MTBF of the RDC comes from a booklet published by a RDC manufacturer [28]. The HF exists in a specific line replaceable unit (LRU), so these data vary with different LRU. The MTBF of the HF refers to the devices in the display system designed by the China National Aeronautical Radio Electronics Research Institute (CARERI). Components fail instantaneously without any common cause effect. Since the sensors do not belong to the IMA system, their failures are not calculated in Section 3.3.

**Table 1.** Failure rate of components in the IMA system.

| Component | Mean Time Between Failure (MTBF) | Failure Rate per Flight Hour |
|---|---|---|
| Sensor (SEN) | 20,000 h | $5.00 \times 10^{-5}$ |
| Switch (SW) | 100,000 h | $1.00 \times 10^{-5}$ |
| Remote data concentrator (RDC) | 14,000 h | $7.14 \times 10^{-5}$ |
| Hosted function (HF) | 16,000 h | $6.25 \times 10^{-5}$ |
| General processing module (GPM) | 50,000 h | $2.00 \times 10^{-5}$ |

In this paper, the meaning of the failure condition is similar to the functional failure mode. According to their severity, failure conditions can be classified into catastrophic, hazardous, major, minor, and of no safety effect. On the basis of the design experience of civil aircraft projects, a safety-critical function is defined in the model: at least one HF can get both sets of sensor data

processed by the GPM. In other words, a hazardous failure condition defined and denoted as LOSS_SEN_HF. LOSS_SEN_HF means the crew, both HFs, cannot get either set of sensor data processed by the GPM. Development assurance level (DAL) is defined in aerospace recommended practice (ARP) 4754A [6] and the above function should satisfy with DAL B, which means this failure condition may cause the hazardous effect and its failure rate must be lower than $10^{-7}$ per flight hour [29].

### 3.3. Results

We added observers and assertions for a special failure condition based on the alt file generated by xml2alt4ima tool. Then, we ran its program in OpenAltaRica tool, the aim of which was to develop a complete set of tools for the high-level modeling language AltaRica 3.0 [30]. Then, we generated a fault tree in open probabilistic safety assessment (OPSA) format from AltaRica 3.0 model as below.

As shown in Figure 3, the top event of the fault tree is LOSS_SEN_HF, and the basic events are the failures of these components. The size of the generated OPSA file was 8318 KB. There were thousands of automatic defined gates, including all the combinations of different basic events.

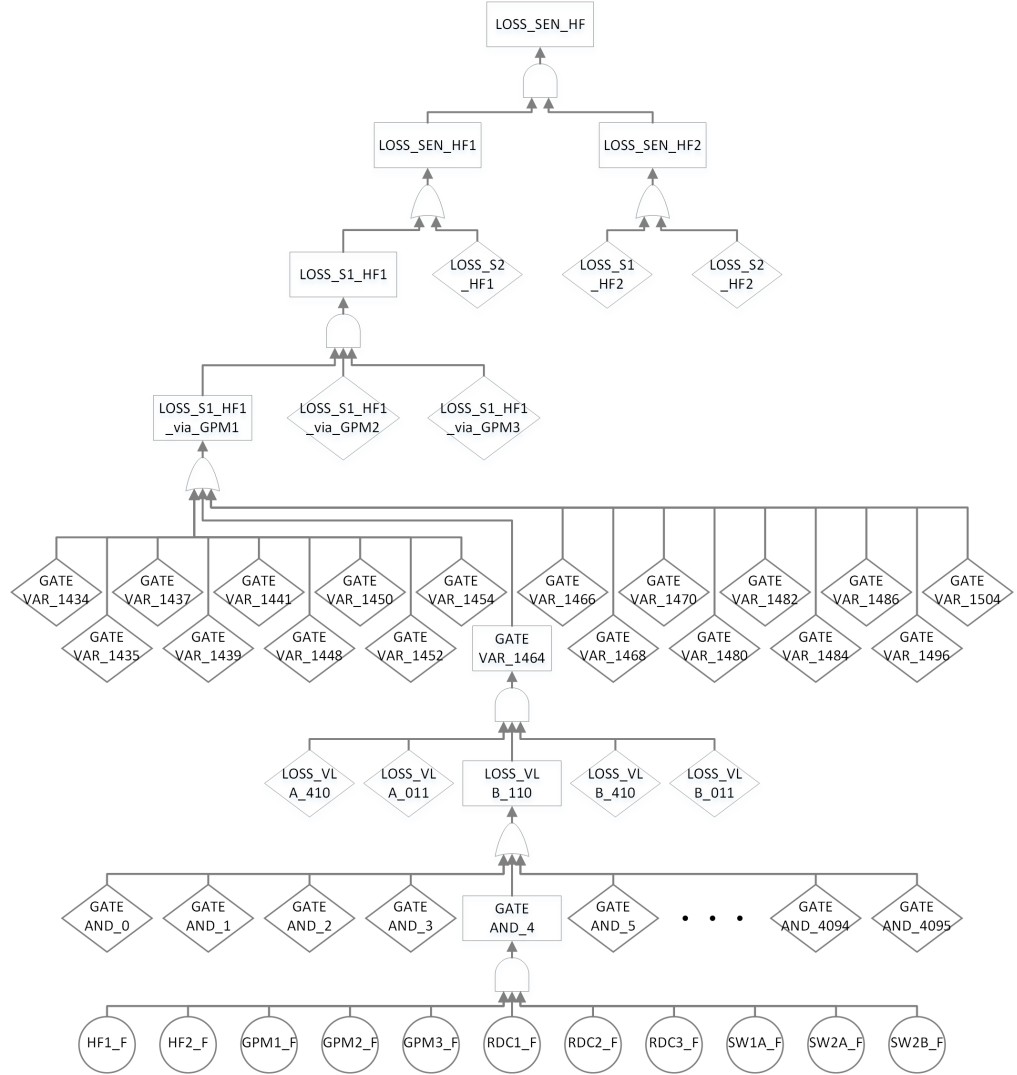

**Figure 3.** The fault tree of the defined failure condition

In addition, we modified the file (XFTA.xml) to meet with the failure conditions defined above [31,32]. Finally, we got the minimal cut set of the failure condition, as shown in Table 2.

**Table 2.** Minimal cut set of LOSS_SEN_HF in the IMA system.

| Rank | Minimal Cut Set | Probability |
|------|-----------------|-------------|
| 1 | rdc1_f, rdc3_f | $5.10 \times 10^{-9}$ |
| 2 | rdc2_f, rdc3_f | $5.10 \times 10^{-9}$ |
| 3 | hf1_f, hf2_f | $3.91 \times 10^{-9}$ |
| 4 | gpm1_f, gpm2_f, gpm3_f | $8.00 \times 10^{-15}$ |
| 5 | rdc1_f, sw2A_f, sw2B_f | $7.14 \times 10^{-15}$ |
| 6 | rdc2_f, sw2A_f, sw2B_f | $7.14 \times 10^{-15}$ |
| 7 | rdc3_f, sw1A_f, sw1B_f | $7.14 \times 10^{-15}$ |
| 8 | hf1_f, sw2A_f, sw2B_f | $6.25 \times 10^{-15}$ |
| 9 | hf2_f, sw1A_f, sw1B_f | $6.25 \times 10^{-15}$ |
| 10 | gpm3_f, sw1A_f, sw1B_f | $2.00 \times 10^{-15}$ |
| 11 | hf2_f, rdc1_f, sw1A_f, sw2B_f | $4.46 \times 10^{-19}$ |
| ⋮ | ⋮ | ⋮ |

Since switches are configured as redundant devices, the IMA configure tool denotes sw1A and sw1B to represent switches with the same location. There are hundreds of minimal cut sets generated by OpenAltaRica, while three second-order cut sets and seven third-order cut sets make up the majority of the top event.

The probability of LOSS_SEN_HF per flight hour is $1.41022 \times 10^{-8}$, which complies with the safety requirements.

Besides, we used other assessment methods to verify the proposed method. We utilized Simfia (software developed by APSYS) to build the model of the above system, generate the fault tree, and calculate the availability. The model and the fault tree of Simfia are shown in Figure 4.

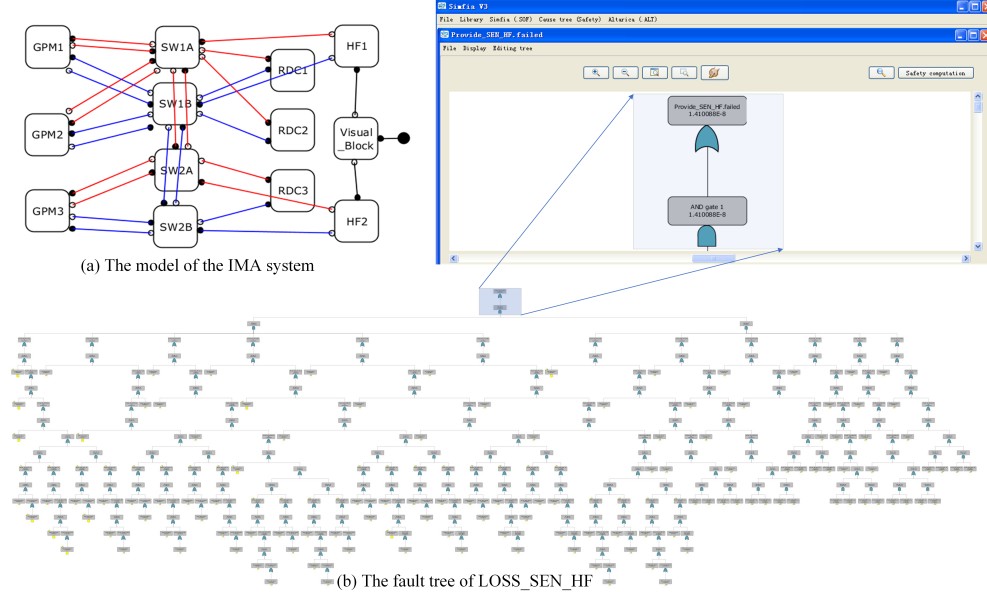

(a) The model of the IMA system

(b) The fault tree of LOSS_SEN_HF

**Figure 4.** The model and the fault tree of Simfia. (**a**) Description of the IMA model. (**b**) The fault tree generated by the Simfia.

The availabilities of the above two methods were a little different ($1.41088 \times 10^{-8}$ calculated by Simfia), and the deviation comes from the computational accuracy of Simfia. We also used the Monte Carlo simulation to verify the proposed method. The failure condition "LOSS_SEN_HF" occurred 141 times over 10,000,000,000 simulation runs. The result of the Monte Carlo simulation is approximately the same with the above two methods. In summary, the Monte Carlo simulation proves the correctness of our method.

### 3.4. Optimization of the IMA System

The GPM in the IMA system is always configured with many applications, thus it is capable of supporting numerous upper functions of differing critical levels. In this paper, we only define one function: the HF obtains sensor data from the RDC processed by GPMs. All devices are redundant in improving the availability of the above function.

As shown in Table 2, the cut set rank first and second are the combination failures of the RDCs, which means that the failures of both data sources can lead to a top event. The cut set rank third, meaning the failures of both data destinations, can lead to a top event as well. The third-order cut set is a combination of different types of components, excluding the cut set rank fourth, which is the failure set of all GPMs. As is well known, three redundant devices failing per flight hour is an event with a small probability, unless these devices are designed with the same unknown bug.

For the function defined in this paper, it seems that there is no need to use three GPMs to satisfy the safety requirement, which should be researched further. Then, we tried to modify the logical architecture with only two GPMs, and generate the alt file referring to the prior subsection. The probability of LOSS_SEN_HF per flight hour with two GPMs is shown in Table 3.

**Table 3.** Probability of LOSS_SEN_HF in the IMA system with two GPMs.

| Configuration | Probability of LOSS_SEN_HF |
|---|---|
| without GPM1 | $1.45022 \times 10^{-8}$ |
| without GPM2 | $1.45022 \times 10^{-8}$ |
| without GPM3 | $1.46022 \times 10^{-8}$ |

Note that the original IMA system includes three GPMs: GPM1, GPM2, and GPM3. Configuration of "without GPM1" denotes the current defined function in the IMA system model only employing GPM2 and GPM3.

Table 3 shows that the probability of LOSS_SEN_HF in the IMA system without GPM1 or GPM2 is lower than that without GPM3. This is due to the fact that in the latter configuration there exists another second-order cut set "sw1A_f, sw1B_f", more than in the first two configurations. As Figure 2 shows, switches are designed to connect with part of the GPMs to balance the communication load. In the above model, it can be concluded that the load balancing designation can reduce the risk of common cause failure, especially for same zone risk.

### 3.5. Efficiency of Safety Assessment Process

The advantage and disadvantage of the three safety assessment methods are analyzed in this section. The first one uses the proposed method based on the MBSA, the second one uses the safety assessment tool (Simfia) to build the safety model manually, and the third one uses the traditional tool (PTC Windchill Quality Solutions, also known as Relex) to build the fault tree based on engineering experience directly. The main steps of obtaining the availability of the failure condition with the different methods are summarized in Table 4. "Auto" means the corresponding step can be operated by software, while "manually" means that the step should be done by a safety engineer, and the context after [source] of each [auto] signifies the data source of corresponding step. It is clear to see that the MBSA requires less manual operations.

With the help of the senior safety engineers in CARERI who have five years experience in using both Simfia and Relex, we tried to analyze the availability of different IMA cases for analyzing the efficiency of the above three methods. All our experiments were conducted on a single core of a 2.7 GHz Intel Core2Duo processor with 2 GB RAM running on Windows XP. The statistical data are shown in Table 5.

**Table 4.** Main steps of obtaining the availability of failure condition.

| Main Steps | MBSA | Simfia | Relex |
|---|---|---|---|
| 1. Definition of components | [auto] Define the ComponentMode by xml2alt4ima [source] system model (xml file) | [manually] Create the blocks as the components of system | [manually] Define the basic event of the fault tree |
| 2. Configuration of components | [auto] Get the failure rate for every component; Define the event with corresponding failure rate by xml2alt4ima [source] system model (xml file) | [manually] Set the failure rate for every block; Create the connector to link the blocks; Set the state types of IN/OUT port of each connector | [manually] Set the failure rate for every basic event; Define the gate; Set the gate type |
| 3. Modeling of logical causes | [auto] Add the transition as the relationship between Visual Links and Components by xml2alt4ima [source] system model (xml file) | [manually] Set the logical causes of every state type of each OUT connector | [manually] Connect the gate with other gates and basic events |
| 4. Definition of failure condition | [manually] Add an observer for the failure condition | [manually] Define a new block for the failure condition | [manually] Define the gate as the top event of the fault tree |
| 5. Configuration of failure condition | [manually] Build the assertion as the relationship between Failure condition and Visual Links | [manually] Create the connector to link the new blocks; Set the state types of IN/OUT port of each connector; Set the logical causes of every state type of each OUT connector | [manually] Connect the top event gate with other gates and basic events |
| 6. Obtain the fault tree | [auto] Generated by OpenAltaRica [source] safety model (alt file) | [auto] Generate the fault tree [source] safety model | [manually] Build the fault tree from top to down |
| 7. Obtain the cut set of failure condition | [auto] Obtain the cut set of failure condition [source] fault tree (opsa file) | [auto] Obtain the cut set of failure condition [source] safety model | [auto] Obtain the cut set of failure condition [source] fault tree |
| 8. Calculate the availability | [auto] Calculate the availability [source] fault tree (opsa file) | [auto] Calculate the availability [source] safety model | [auto] Calculate the availability [source] fault tree |

**Table 5.** Comparison of different methods for the IMA cases.

| | Case 1 [a] | | | Case 2 [b] | | | Case 3 [c] | | |
|---|---|---|---|---|---|---|---|---|---|
| | **MBSA** | **Simfia** | **Relex** | **MBSA** | **Simfia** | **Relex** | **MBSA** | **Simfia** | **Relex** |
| Quantity of components in model | 10 Components | | | 12 Components | | | 15 Components | | |
| Time for modeling | ≈1 h | ≈4 h | ≈6 h | ≈1 h | ≈6 h | ≈10 h | ≈1 h | ≈8 h | ≈12 h |
| Time for generating fault tree | <1 min | <10 s | none | <10 min | < 10 s | none | <1 h | <1 min | none |
| Quantity of gates and basic events [d] | 1254 | 167 | 84 | 5828 | 232 | 115 | 45,844 | 282 | 136 |
| Time for calculating the cut set | <1 min | <10 s | <10 s | 1 min | <10 s | <10 s | 1 min | <10 s | <10 s |
| Time for remodeling [e] | ≈40 min | ≈1 h | ≈3 h | ≈1 h | ≈2 h | ≈5 h | ≈1 h | ≈3 h | ≈6 h |

[a] The IMA system with two HFs, two GPMs, four switches, and two RDCs. [b] The IMA system in the prior subsection with two HFs, three GPMs, four switches, and three RDCs. [c] The IMA system with four HFs, three GPMs, four Switches, and four RDCs. [d] The fault tree contains many useless gates generated by the OpenAltaRica and Simfia. The repeated basic event of the fault tree was only counted once. [e] The modification mainly refers to the change of logical architecture of the IMA model.

Table 4 shows that our proposed method based on the MBSA costs less time to model than the traditional method or Simfia. When the system changes, our proposed method can be changed more efficiently than the other two methods. However, the quantity of gates and basic events of the fault tree generated by OpenAltaRica is larger than that of the other two methods, and the time for calculating the cut set of our proposed method is longer. In addition, different analysts have different styles to define the gate of the fault tree, which makes it difficult for other analysts to understand the fault tree generated by Relex. When utilizing Simfia to analyze the availability, analysts need to remodel the system according to their comprehension, which is inefficient and prone to error. On the contrary, our proposed method enables the safety analyst to devote their time to the safety analysis and designation advice rather than on duplicate work.

## 4. Conclusions

IMA is recommended because of its high utility as regards resources and hierarchical architecture, as well as its ease of use for the engineer. However, the traditional availability assessment of the IMA system with the feature of the redundant AFDX network is time-consuming and error-prone. For a safety-critical system, the common way to analyze the availability is through modeling the fault tree based on engineering experience. In this paper, we propose an availability assessment method for the IMA system based on MBSA using AltaRica 3.0 and implement a tool to generate an alt file based on the configuration xml file of the IMA system. In this way, the availability assessment becomes faster and can be modified effectively according to the change of system. Taking a typical IMA system model as an example, the results indicate that the application in the IMA system satisfies the safety requirements. In addition, we also find that the load balancing designation of the IMA system is advantageous in reducing the risk of common cause failure. Our method can also be used in the availability assessment of a safety-critical system with hierarchical architecture with a functional–logical–physical layer.

In the future, we will research the safety analysis of the IMA system in the following two aspects: On the one hand, considering that different GPMs can process respective functions, it is necessary to study the feature of the fault propagation process in the IMA system; on the other hand, it is of great importance to study a process which can handle the availability assessment considering errors or the confusing of basic components and functions, rather than the loss of these modules.

**Author Contributions:** Conceptualization, H.D.; Data curation, Z.Z.; Formal analysis, G.W.; Funding acquisition, G.W.; Investigation, Z.Z.; Methodology, H.D., G.W., Z.Z., Y.L. and M.W.; Project administration, Q.G.; Resources, Q.G.; Software, H.D.; Validation, Q.G.; Writing—original draft, H.D.; Writing—review & editing, Y.L. and M.W.

**Funding:** This research was funded by National Key Basic Research Program of China grant number 2014CB744900.

**Acknowledgments:** The authors would like to thank Hao Rong for his technical assistance in the construction of the case model, Zilong He, Ning Guo and Ruoxing Mei for their advice and assistance in proofreading the article.

**Conflicts of Interest:** The authors declare no conflict of interest.

## Abbreviations

The following abbreviations are used in this manuscript:

| | |
|---|---|
| IMA | Integrated Modular Avionics |
| AFDX | Avionics Full Duplex Switched Ethernet |
| VL | Visual Link |
| FTA | Fault Tree Analysis |
| MBSA | Module Based Safety Analysis |
| ARP | Aerospace Recommended Practice |
| GTS | Guarded Transition Systems |
| FPM | Failure Propagation Model |
| CARERI | China National Aeronautical Radio Electronics Research Institute |

ICD        Interface Control Document
GPM        General Processing Module
RDC        Remote Data Concentrator
HF         Hosted Function
SW         Switch
SEN        Sensor
BU         Backup
ARINC      Aeronautical Radio Inc.
MTBF       Mean Time Between Failure
LRU        Line Replaceable Unit
DAL        Development Assurance Level
KB         Kilo Byte
OPSA       Open Probabilistic Safety Assessment

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
