# Peer review of "Availability Assessment of IMA System Based on Model-Based Safety Analysis Using AltaRica 3.0"

_processes, doi:10.3390/pr7020117_

Round 1
Reviewer 1 Report
This article is definitely interesting.
As a minor improvement, I think it would good to give more details about the XML format used as starting point of the analysis (in terms of content).
Also, it may be interesting to add a small discussion about the chosen assessment tool (compilation into fault tree). Would it be of interest for example to use Monte-Carlo simulation?
Author Response
Dear Reviewer,
We are very grateful to the editors and the referees for evaluating our manuscript entitled “Availability Assessment of IMA System based on Model-Based Safety Analysis using AltaRica 3.0” and for the comments and suggestions. According to the suggestions, we have carefully revised the manuscript. The main corrections can be found in the revised manuscript in RED color. In the attachment, we quote the comments from the reviewers, and present our response and revision. We look forward to hearing from you again soon.
Sincerely yours,
Haiyong Dong, On behalf of all authors

Reviewer 2 Report
The proposed publication named "Availability Assessment of IMA System based on Model-Based Safety Analysis using AltaRica 3.0" presents a methodology to assess a specific critical avionic system: the IMA system. This methodology is presented as based on MBSA and uses the AltaRica 3.0 modeling language. It is meanly explained by means of an IMA system example.
It is not clear how the methodology works. The figure 1 and its associated explanation presents three different elements an XML file, a model of the IAM system and a (generated) AltaRica file. First, the figure and explanations focus more on the files (xml and alt), which seems not relevant. they must focus on concepts handled by these files. Second it is not clear how the IMA system is designed: which tool. Finally the use of Matlab to develop the tool xml2alt4ima must be justified. The presented algorithm 1 must be improved: for example the instruction "Make sure the domain of ComponetMode exists, if not, add it" can be divided onto several instructions and the algorithm becomes more complex.
The part concerning the case study is relatively clear. Nevertheless the final part according to the use of Simfia, and explanations on the time to design the model, is not relevant. In fact, firstly the proposed methodology does not take into account the time to build the IMA system model and the xml file. Furthermore, system and safety models are generally not equals: there is not a one to one correspondence between part of the system model and part of the safety model. For example some elements are simplified into the safety models.
Finally the part concerning the comparison between the different methods is not relevant. In fact, take into account the time to design the model as a comparison criteria is not serious. It depends on the user experience of the designer with the tool.
This publication is poorly written with several mistakes.
Author Response

(The authors gave the same response as above.)

Round 2
Reviewer 2 Report
Dear Authors, Thank you very much for the improvement of your document.
Corrections are clear and increase the quality of the document.
Nevertheless there are still two elements that can/must be improved.
The first one concerns the methodology presented by Figure 1. The first review questioned the fact that this methodology is presented from a ‘file’ point of view, meaning that explanations talk about files (xml), whereas it must be presented from a ‘concept’ point of view. Xml files are not made to be handled by human. It is only an efficient support to exchange data between (software) tools. Even the second version of the document presents more precisely the way to define this file, it still focuses on this file. Furthermore Figure 1 presents the elements handled and recorded into the xml file, but not with the xml syntax. A sample of a generated xml file is also added, but it does not clarify the explanations. We suggest to authors to remove this sample of the generated xml file and let all the other explanations.
The second one concerns the comparisons between the different methodologies (Part 3.5). Once again, the comparison considers features that cannot be compared directly. For example Line 2 of Table 4 considers the configuration of components that is done automatically with the MBSA approach and manually with Simfia and Relex. It is not true, or more precisely, it forgets that for example failure rates must be set by someone during the overall design process, even if it is not during the safety analysis.
Author Response
Dear Reviewer,
We are very grateful to you for evaluating our revised manuscript entitled “Availability Assessment of IMA System based on Model-Based Safety Analysis using AltaRica 3.0” and for the comments and suggestions. According to the suggestions, we have carefully revised the manuscript. The main corrections can be found in the revised manuscript in RED color. In the attachment, we quote the comments and present our response and revision. We look forward to hearing from you again soon.
Sincerely yours,
Haiyong Dong, On behalf of all authors
